# Estimating the cost impact of atrial fibrillation using a prospective cohort study and population-based controls

Helena Aebersold [iD] ,[1] Fabienne Foster-Witassek,[1] Miquel Serra-Burriel,[1] Beat Brüngger [iD] ,[2] Stefanie Aeschbacher,[3,4] Jürg-Hans Beer,[5,6] Eva Blozik,[7] Manuel Blum,[8,9] Leo Bonati,[10,11] David Conen,[12] Giulio Conte,[13] Stefan Felder,[14] Carola Huber [iD] ,[2] Michael Kuehne,[3,4] Giorgio Moschovitis [iD] ,[15] Andreas Mueller,[16] Rebecca E Paladini,[3,4] Tobias Reichlin,[17] Nicolas Rodondi,[8,9] Anne Springer,[3,4] Annina Stauber,[16] Christian Sticherling,[3,4] Thomas Szucs,[18] Stefan Osswald,[3,4] Matthias Schwenkglenks,[1,18] On behalf of the Swiss-AF Investigators

For numbered affiliations see end of article.

**Correspondence to**
Dr Helena Aebersold;
helena.aebersold@uzh.ch

## ABSTRACT

**Aims** Atrial fibrillation (AF) costs are expected to be substantial, but cost comparisons with the general population are scarce. Using data from the prospective Swiss-AF cohort study and population-based controls, we estimated the impact of AF on direct healthcare costs from the Swiss statutory health insurance perspective.

**Methods** Swiss-AF patients, enrolled from 2014 to 2017, had documented, prevalent AF. We analysed 5 years of follow-up, where clinical data, and health insurance claims in 42% of the patients were collected on a yearly basis. Controls from a health insurance claims database were matched for demographics and region. The cost impact of AF was estimated using five different methods: (1) ordinary least square regression (OLS), (2) OLS-based two-part modelling, (3) generalised linear model-based two-part modelling, (4) 1:1 nearest neighbour propensity score matching and (5) a cost adjudication algorithm using Swiss-AF data non-comparatively and considering clinical data. Cost of illness at the Swiss national level was modelled using obtained cost estimates, prevalence from the Global Burden of Disease Project, and Swiss population data.

**Results** The 1024 Swiss-AF patients with available claims data were compared with 16 556 controls without known AF. AF patients accrued CHF5600 (EUR5091) of AF-related direct healthcare costs per year, in addition to non-AF-related healthcare costs of CHF11 100 (EUR10 091) per year accrued by AF patients and controls. All five methods yielded comparable results. AF-related costs at the national level were estimated to amount to 1% of Swiss healthcare expenditure.

**Conclusions** We robustly found direct medical costs of AF patients were 50% higher than those of population-based controls. Such information on the incremental cost burden of AF may support healthcare capacity planning.

## INTRODUCTION

Atrial fibrillation (AF) is the most common form of serious arrhythmia worldwide, and a major cause of stroke and heart failure.

## STRENGTHS AND LIMITATIONS OF THIS STUDY

⇒ This study used 5 years of follow-up data from a large prospective cohort of prevalent atrial fibrillation (AF) patients.

⇒ The direct medical cost impact of AF was assessed by comparison with population-based controls drawn from a large health insurance database.

⇒ Several regression-based and propensity score-based methods were used to judge robustness and AF costs were also assessed using a non-comparative approach.

⇒ The cohort of AF patients may not be fully representative of all AF patients.

⇒ A limited degree of residual presence of AF in the control population cannot be ruled out.

More than 11 million people live with AF in Europe.[1] [2] Given demographic ageing, Europe is expected to face a larger increase in AF prevalence by 2050 than any other region globally.[1]

Several studies on cost of illness of AF have estimated costs at the patient or nationwide levels. Direct healthcare costs per patient were estimated to range from EUR2315 to EUR3307 annually in Europe,[3–6] and from US$6410 to US$8705 in the USA.[7] [8] At the national level, direct costs of AF in Europe may range from EUR660 to EUR2548 million;,[9–12] in the USA, they were estimated at around US$6 billion[8] [13] These costs are substantial, accounting for 0.28%–1.7% of the national health expenditures of these countries.[12] [14–16]

So far, most attempts assessing the cost impact of AF remained descriptive. To our knowledge, only two studies[8] [16] compared costs between AF patients and a control population. Even less evidence is available for cost

changes since 2010, as most cost-of-illness studies rely on data collected earlier.

We used a recent real-world dataset from a large prospective cohort study of AF patients to assess the yearly cost impact of AF. Comparing with a population-based control sample, direct healthcare costs of AF were estimated at the patient level and transferred to the national level. Results were compared with estimates resulting from an adjudication algorithm only using the cohort data in a non-comparative approach.

## METHODS
### Study design and data sources
Swiss-AF is a large, ongoing prospective observational cohort study across 14 clinical centres in Switzerland, investigating AF-related cognition, complications and economic aspects. Patients were enrolled between 2014 and 2017 if they had a history of documented AF and were older than 65 years; 228 patients were enrolled aged 45–64 to enhance the study of socioeconomic aspects. A data cut of 2014–2020 was used in this analysis. The detailed study setup has been published earlier.[17]

Alongside clinical data, health economic data were collected. These included medical resource use at the study centres, and health insurance claims from four cooperating health insurers covering 42% of the study sample. In Switzerland, health insurance is compulsory and offered to anyone, covering inpatient and outpatient services. The benefit package is uniform across the country and defined by law.

To assess the cost impact of AF, a population-based reference sample was provided by Helsana, an insurer covering about 15% of the Swiss population. Helsana enrolees were eligible for the reference sample if they were not Swiss-AF patients, were in the same age range as the Swiss-AF population, and had statutory health insurance claims data available for a period equivalent to the one available for Swiss-AF patients. For the reference sample a subset of 19 002 patients was randomly selected, frequency matched to the Swiss-AF patients by age, gender and geographical region (online supplemental table S1). To ensure similar observation times, start dates for the controls were randomly assigned using the distribution of Swiss-AF enrolment dates. Sensitivity analyses with different starting and ending dates were run without altering the results significantly. Individuals within the reference sample could have AF, as Swiss claims data do not have direct diagnosis information for outpatient services. Hence, a categorisation algorithm (online supplemental table S2) was developed together with clinicians from the Swiss-AF centres to distinguish such persons. Using codes from the International Classification of Diseases 10th Revision,[18] the Swiss diagnosis-related group-based (SwissDRG)[19] flat fee reimbursement system for inpatient episodes, the Swiss invasive medical procedures catalogue (CHOP),[20] the anatomical therapeutic chemical classification (ATC) of medicines,[21]

and the national tariff for outpatient physician services (Tarmed),[22] three categories resulted: 'AF likely', 'AF possible' and 'AF not obvious'. We assigned the category of 'AF likely' to patients with a very high probability of having AF, as most codes were hospitalisation based. Persons categorised as 'AF possible' had codes possibly but not clearly allocable to AF. All other patients were classified as 'AF not obvious' and considered as controls (figure 1).

Equivalent claims data were available for the Swiss-AF and control patients, reflecting all claims for reimbursement by the Swiss statutory health insurance. The claims data included detailed information on outpatient services and drugs, and less detailed information on inpatient services based on SwissDRG.[19] Given the absence of clinical data for the control sample, the presence of major chronic morbidities was approximated, uniformly for Swiss-AF patients and controls, based on outpatient drug claims, using the pharmaceutical cost groups (PCG) approach.[23]

### Outcome measures
Our main outcome of interest was the AF-induced part of direct medical healthcare costs from the perspective of the Swiss statutory health insurance. To assess the cost impact of AF, the Swiss-AF patients were compared with the population-based controls, using different multivariable regression methods: (1) ordinary least square regression (OLS), (2) OLS-based two-part modelling, (3) generalised linear model (GLM)-based two-part modelling and (4) 1:1 nearest neighbour propensity score matching. Furthermore, (5) estimates were compared with AF costs estimated using a previously developed adjudication algorithm.[24] In brief, the AF-adjudication algorithm combined clinical event data collected in Swiss-AF with health insurance claims, adjudicating each cost component as AF-related or non-AF related. We distinguished between total, outpatient and inpatient costs. All cost calculations considered individual start dates and follow-up times and were aggregated to a yearly level. Given the relative stability of prices over the observation period, costs were taken as recorded in the health insurance database. To facilitate comparison with other countries, main cost results are presented in Euros (EUR) in addition to Swiss francs (CHF), based on an exchange rate (averaged 2014–2020) of EUR1.0=CHF1.1. Individual follow-up times were censored at 5 years after the start date due to the small number of longer follow-up periods available.

### Covariates
Covariates available for both the Swiss-AF and control population included the following types: first, patient characteristics: age, sex and area of residence (greater regions of Switzerland); second, PCGs as proxies for comorbidities: acid-related disorders (ie, gastro-oesophageal reflux disease), bone diseases, cancer, dementia, epilepsy, respiratory illness, rheumatic conditions, glaucoma, gout, iron

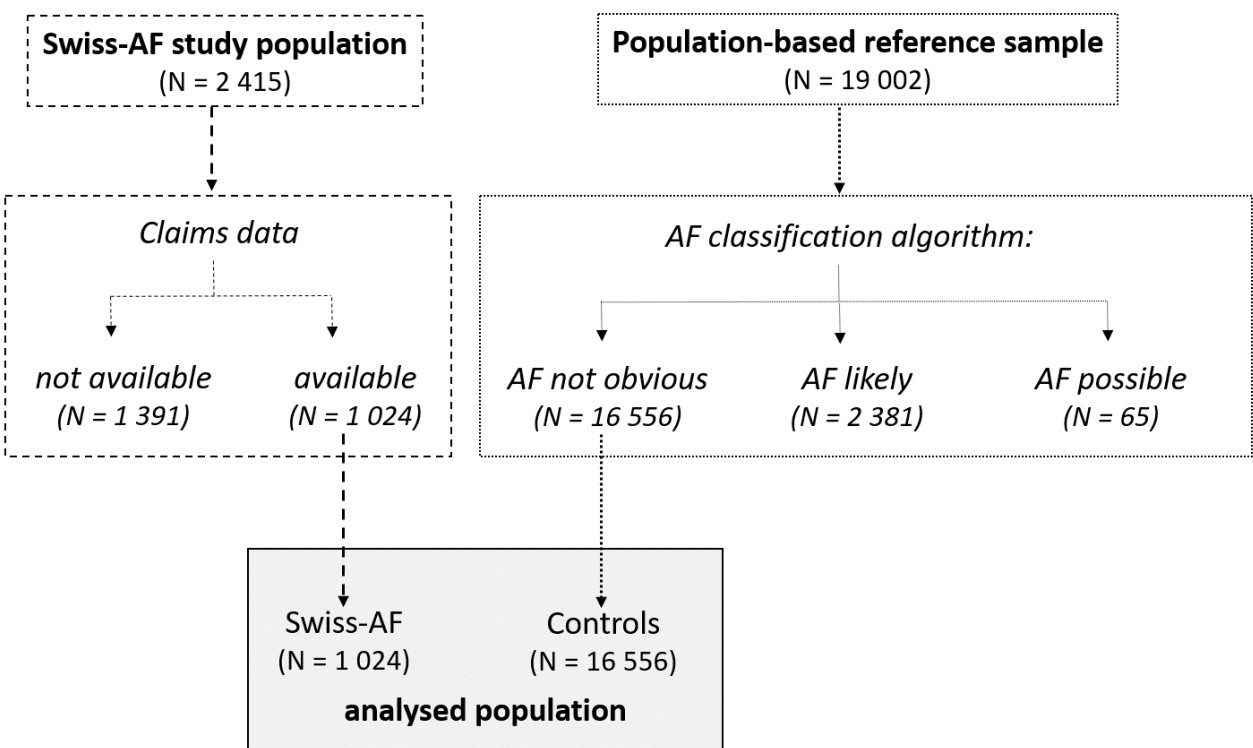

**Figure 1** Flow chart: patient and cohort selection. Individuals classified as "AF likely" had an ICD10 (e.g. I48) or CHOP code indicative of AF. Those with "AF possible" had ICD10 or Swiss-DRG codes which may relate to AF (e.g. ICD10: I49, DRG: F50D). All others from the reference sample were classified as "AF not obvious" and considered as controls. ICD, international classification of diseases (10th revision); CHOP, Swiss invasive medical procedures catalogue; DRG, diagnosis related group; AF, atrial fibrillation.

deficiency, chronic pain, psychiatric diseases, use of anti-psychotic drugs, thyroid disease and other rare diseases and third, a year of follow-up. Insurance characteristics were obtained from three of four insurers and considered in a sensitivity analysis.

### Statistical analysis and estimation of AF costs per person

First, the characteristics of the included Swiss-AF and control patients were described with standard methods. Healthcare costs per patient and cost trajectories over time were descriptively analysed for both populations, distinguishing between total, outpatient and inpatient costs. Cost trajectories over time were depicted as line plots not considering missing data points.

Second, the mentioned multivariable regression approaches were pursued to assess the cost impact of AF, using the above-listed covariates as independent variables. All approaches included a time fixed effect for month of observation.

The two-part alternatives to OLS were pursued because healthcare costs are characterised by a significant proportion of zero values and right-skewed distributions of non-zero costs.[25] In the first part of the two-part models, the probability of having any costs in a given year of follow-up was estimated using a logistic model. The same covariates were used in the second part of

the model, estimating the costs conditional on having occurred. Again, OLS was chosen for the second part to achieve direct cost estimates. Alternatively, GLMs with an assumed gamma distribution and logarithmic link function were used in the second part, to better account for the heteroscedasticity typically present in healthcare costs.[26] The cost ratios of the GLM part were converted to marginal effects to enable a direct comparison with the OLS-based results. Mean annual costs were finally calculated by multiplying the predicted values of both modelling parts.[27] To estimate the marginal cost impact of AF, all patients were assumed to have AF, or not to have AF. Both sets of predicted values were calculated, and the difference was interpreted as the cost impact of AF.[28] A further analysis was run by estimating the AF costs with propensity score matching, using a 1:1 nearest neighbour approach. Given the characteristics of the data, the GLM-based two-part modelling approach was considered theoretically most suitable, and the corresponding results were treated as primary.

Third, the different regression-based estimates of AF costs were compared with the estimates of AF costs resulting from applying the AF adjudication algorithm to the Swiss-AF patients' claims data.[24]

## AF costs at the national level

Fourth, cost of illness of AF for Switzerland was roughly approximated as total costs per year, and costs per inhabitant and year, for the time period 2000–2019. Mean annual AF-related costs were taken from the GLM-based two-part model and assumed to follow the trend of healthcare expenditures in Switzerland for the period (index 2019=100%). AF prevalence was taken from the database of the Global Burden of Disease Project for the Swiss population older than 30.[2] For cost calculations per capita, the Swiss population size was used with no age restriction, obtained from the Swiss Federal Statistical Office.[29]

All analyses were conducted by using R V.3.6.3.

## Patient and public involvement

Patients or the public were not involved in the design, or conduct, or reporting, or dissemination plans of our research.

## RESULTS
### Patient population

Figure 1 shows the cohort selection. Of 2415 Swiss-AF patients, 1024 (42.4%) had claims data available and were included in the analysis (patients without available claims data showed similar characteristics).[24] In the population-based reference sample, 16 556 individuals were classified as 'AF not obvious' and included as controls. Baseline characteristics by cohort are shown in table 1. Online supplemental figure S1 provides details on the numbers of patients at risk, cumulative numbers of events, the development of costs and Kaplan-Meier survival estimates across the full observation period 2014–2020 by cohort.

### Healthcare costs over time

The evolution of mean annual costs by cohort and cost component is depicted in figure 2 (details in online supplemental table S3 and figure S2). The unadjusted average total cost per patient and year amounted to CHF19037 (EUR17 306) for Swiss-AF patients, around 1.7-fold more than for control patients. In both cohorts, inpatient and outpatient costs each contributed half of the total costs on average.

### AF-related and non-AF-related healthcare costs

Table 2 compares the model-based estimated differences in healthcare costs between AF patients and controls, interpreted as AF-related costs. Details for each model are in online supplemental tables S4–S7. All estimates of AF-related costs were in a similar range. The GLM-based two-part model yielded total AF costs of CHF5588 (EUR5080) annually, while outpatient costs were CHF1425 (EUR1295), and inpatient costs CHF2779 (EUR2526).

Figure 3 compares the estimates of AF-related costs from the GLM- and OLS-based two-part models with the estimates for the Swiss-AF patients based on the

AF-adjudication algorithm without controls. The estimated AF-related costs were very similar for all three methods, ranging from CHF5187 (OLS-based) to CHF5588 (GLM based), and CHF5679 (adjudication based). AF-related costs from the adjudication algorithm are shown by subgroup, revealing details not available

**Table 1** Baseline characteristics

| | Swiss-AF | Controls | SMD |
|---|---|---|---|
| N | 1024 | 16 556 | |
| **Characteristics** | | | |
| Age mean (SD) | 73.04 (8.17) | 72.64 (8.52) | 0.401 |
| Sex male N (%) | 741 (72.4) | 11 766 (71.1) | 0.145 |
| **Comorbidities (PCG) N (%)** | | | |
| Acid-related disorders | 397 (38.8) | 2802 (17.4) | 0.326 |
| Bone diseases | 44 (4.3) | 644 (4.0) | 0.035 |
| Cancer | 35 (3.4) | 510 (3.2) | 0.067 |
| Cardiovascular | 754 (73.8) | 10 381 (63.7) | 0.402 |
| Dementia | 27 (2.6) | 797 (5.0) | 0.097 |
| Diabetes | 122 (11.9) | 2298 (14.3) | 0.161 |
| Epilepsy | 66 (6.5) | 982 (6.1) | 0.077 |
| Glaucoma | 103 (10.1) | 1634 (10.2) | 0.035 |
| Gout | 96 (9.4) | 935 (5.8) | 0.151 |
| Hyperlipidaemia | 425 (41.6) | 5649 (35.0) | 0.174 |
| Iron deficiency | 66 (6.5) | 567 (3.5) | 0.116 |
| Pain | 386 (37.8) | 2484 (15.4) | 0.347 |
| Psychiatric | 266 (26.0) | 2837 (17.6) | 0.136 |
| Antipsychotic | 16 (1.6) | 878 (5.5) | 0.142 |
| Respiratory | 144 (14.1) | 1915 (11.9) | 0.141 |
| Rheumatic conditions | 406 (39.7) | 3074 (19.1) | 0.309 |
| Thyroid disorders | 87 (8.5) | 908 (5.7) | 0.083 |
| Other rare diseases | 27 (2.6) | 696 (4.4) | 0.107 |
| No of PCGs mean (SD) | 3.39 (2.53) | 2.41 (1.98) | 0.31 |
| **Mother tongue N (%)** | | | 0.108 |
| German | 755 (73.7) | 12 944 (78.2) | |
| French | 141 (13.8) | 1708 (10.3) | |
| Italian | 128 (12.5) | 1904 (11.5) | |
| **Greater region N (%)** | | | 0.182 |
| Zurich | 125 (12.2) | 2083 (12.6) | |
| Lake Geneva Region | 56 (5.5) | 1086 (6.6) | |
| Espace Mitelland | 289 (28.2) | 3702 (22.4) | |
| Northwestern Switzerland | 310 (30.3) | 5990 (36.2) | |
| Eastern Switzerland | 67 (6.5) | 944 (5.7) | |
| Southern Switzerland | 125 (12.2) | 1904 (11.5) | |
| Central Switzerland | 52 (5.1) | 847 (5.1) | |

AF, atrial fibrillation; PCG, pharmaceutical cost groups; SMD, standardised mean difference.

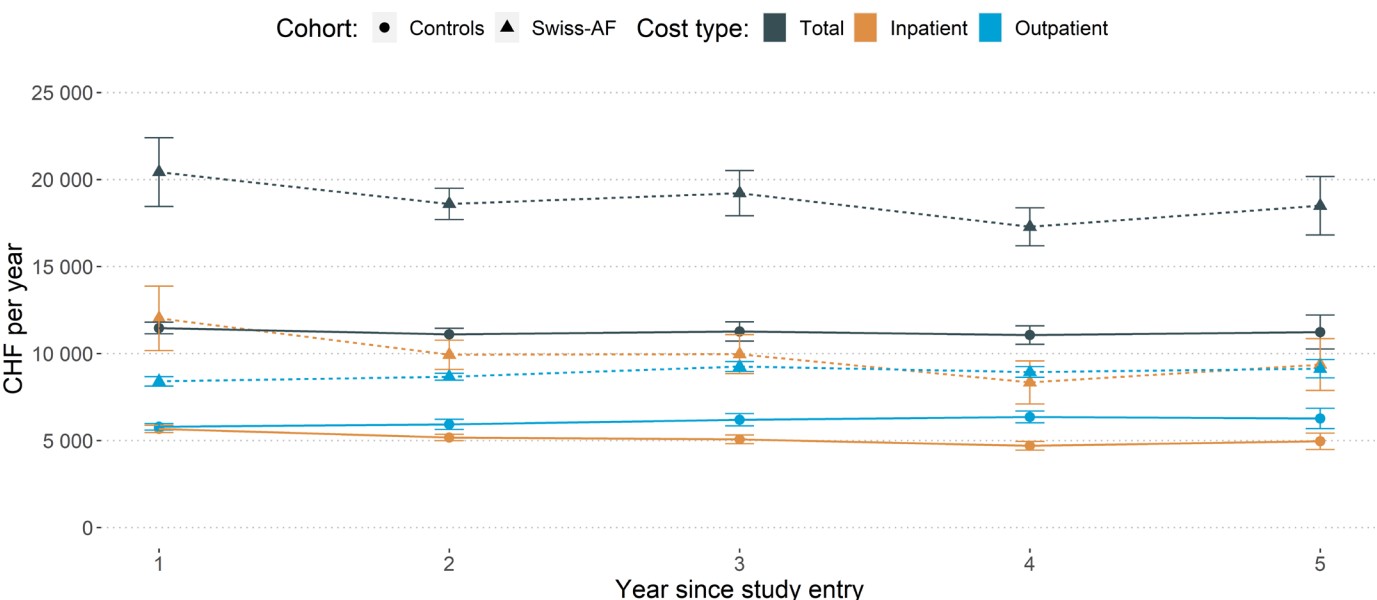

**Figure 2** Healthcare costs over time. Notes: Point estimates are presented by dots, and 95% confidence intervals are presented with error bars. An exchange rate of EUR 1.0 = CHF 1.1 can be used to convert the costs into Euros to facilitate comparison with other countries. AF, atrial fibrillation.

from the regression estimates: AF-treatment costs contributed most to AF-related costs, while the costs of AF-related complications contributed relatively little. Non-AF-related costs induced by diseases other than AF, that is, accrued by the Swiss-AF patients and the controls, were similar across all approaches. They amounted to CHF11 100 (EUR10 091) per year OLS and GLM based, and CHF13 400 (EUR12 182) per year adjudication based.

### Cost of illness in Switzerland

Online supplemental figure S3 shows the estimated evolution of AF-related costs at the Swiss national level, in total and in CHF per inhabitant. Since 2000 the increase

in costs was faster than the prevalence increase of AF in the population. Estimates amounted to CHF700 million (EUR636 million) in 2019, equivalent to about CHF80 per inhabitant. Male patients contributed 1.5 times more to the costs than female patients due to higher prevalence, and most of the costs were accrued in patients older than 70 years (online supplemental figure S4).

### Discussion

This study presents up-to-date evidence of real-world AF-related healthcare costs. To the best of our knowledge, it is the first study comparing AF-related cost estimates using population-based controls with a data-derived

**Table 2** Estimates of difference in healthcare costs between AF patients and controls: comparison of alternative models

| | | Model | | | |
|---|---|---|---|---|---|
| | Dependent variable | Two-part GLM | Two-part OLS | Propensity score matching | OLS |
| Total costs | OR (logistic part) | 1.50 (1.46 to 1.54) | – | – | – |
| | Marginal effect/cost estimate (GLM/OLS part) | 6374 (5609 to 7139) | 5743 (5210 to 6277) | – | – |
| | Combined two part/direct estimate | 5588 | 5187 | 5692 | 5124 |
| Outpatient costs | OR (logistic part) | 1.46 (1.42 to 1.50) | – | – | – |
| | Marginal effect/cost estimate (GLM/OLS part) | 1299 (1097 to 1501) | 1043 (860 to 1226) | – | – |
| | Combined two part/direct estimate | 1425 | 1246 | 1342 | 1124 |
| Inpatient costs | OR (logistic part) | 1.13 (1.08 to 1.17) | – | – | – |
| | Marginal effect/cost estimate (GLM/OLS part) | 35 154 (28 827 to 41 481) | 37 322 (32 916 to 41 728) | – | – |
| | Combined two part/direct estimate | 2779 | 2957 | 4350 | 3999 |

The two-part models used a logistic regression in the first part, and GLM or OLS, respectively, in the second part. Propensity score matching was done 1:1, and OLS refers to a direct (non-two-part) OLS estimate. The brackets show 95% CIs. An exchange rate of EUR1.0=CHF1.1 can be used to convert the costs into Euros to facilitate comparison with other countries.
AF, atrial fibrillation; GLM, generalised linear model; OLS, ordinary least squares regression.

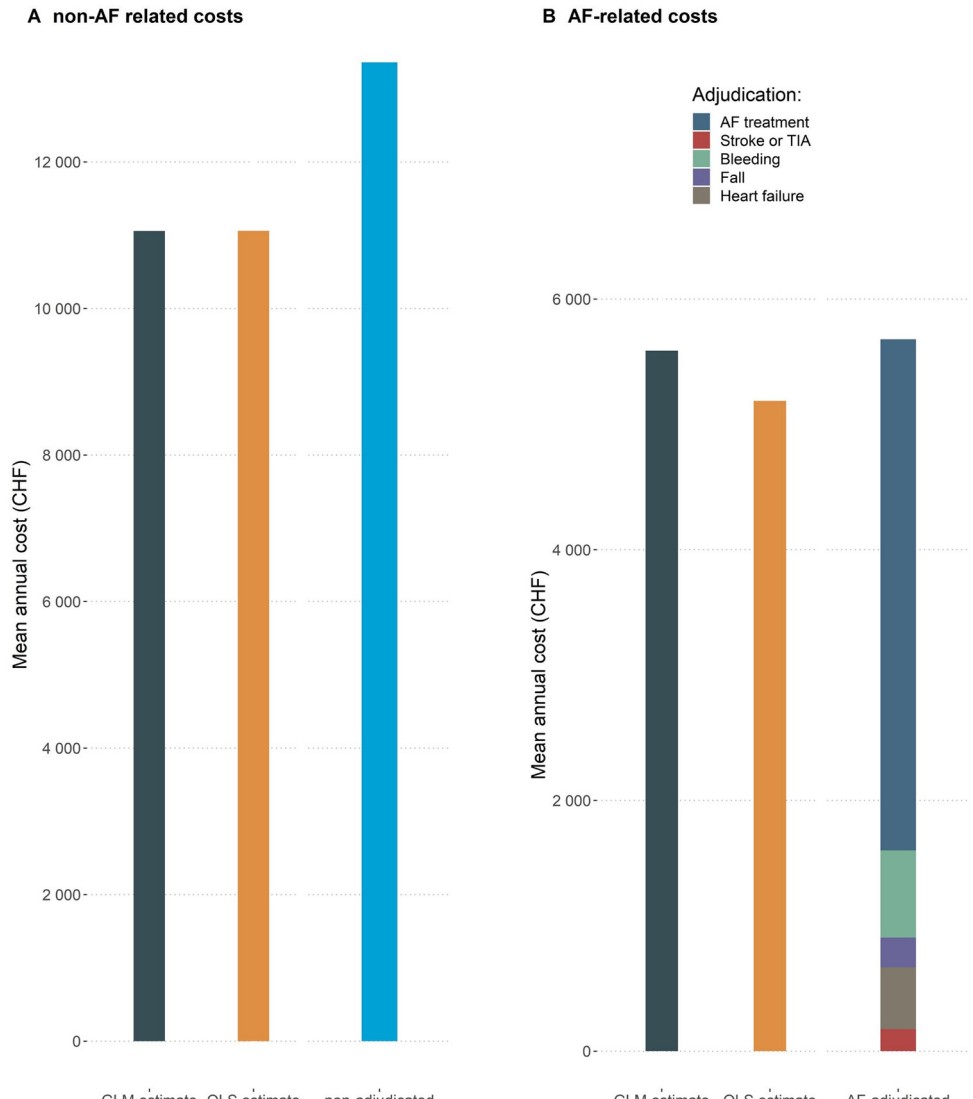

**Figure 3** Estimated non-AF related and AF-related mean annual healthcare costs. Estimates from the GLM-based and OLS-based two part models compared with estimates based on the adjudication algorithm. Notes: An exchange rate of EUR 1.0 = CHF 1.1 can be used to convert the costs into Euros to facilitate comparison with other countries. AF, atrial fibrillation; GLM, generalised linear model; OLS, ordinary least square; TIA, transient ischemic attack.

bottom-up approach to adjudication of AF costs. We obtained similar results for all estimation methods used: mean annual AF-related costs amounted to CHF5600 (EUR5091); indicating roughly 50% higher direct medical costs of Swiss AF patients compared with the population-based controls. At the national level, AF-related costs amounted to CHF700 million (EUR636 million) in 2019, equivalent to about 1% of Swiss healthcare expenditure.

Our estimates of AF-related direct medical costs of CHF5600 annually are consistent with previously published estimates, despite notable differences in study designs and data collection approaches. In Europe, annual direct medical cost estimates at the patient level ranged from EUR2315 to EUR3785 (Spain EUR2315 (2006),[4] Germany EUR2405 (2005),[3] Sweden EUR2787 (2006),[3] Italy EUR3225 (2006),[4] France EUR3307 (2004),[6] Scotland GBP3785 (2015)[5]). After accounting for purchasing power parity (PPP), our estimate for Switzerland is still

somewhat higher, but comparable. As Ringborg et al's study[4] has shown, differences within Europe are notable even after accounting for PPP, reflecting differences in the healthcare systems of the countries. Moreover, Switzerland is known to have a relatively more expensive healthcare system than other European countries.

Transferred to the Swiss national level, direct medical AF costs amounted to CHF700 million in 2019. AF-related cost estimates for European countries ranged from EUR660 to EUR2548 million (Germany EUR660 million (2004),[9] France EUR1942 million (2012),[10] Sweden EUR240 million (2007),[11] UK GBP244 million (1995) to model-based estimates of 2548 million (2020).[12 15] In the USA, AF-related costs were estimated to be around US$6 billion (2008).[8 13] It is difficult to compare the existing cost-of-illness studies due to methodological differences, while differences in their timing and in population size can for example, be captured by expressing

AF-related costs as a share of the gross domestic product (GDP) or total healthcare expenditure in the relevant year. In Switzerland, the estimated AF-related costs amounted to 0.1% of the GDP in 2019, equivalent to roughly 1% of the total healthcare expenditure. This is again comparable with the existing literature. In Portugal, AF-related costs were estimated to be 0.08% of the GDP, including indirect costs but excluding bleeding-related events and services.[30] AF-related cost estimates as a share of healthcare expenditures ranged from 0.28% to 1.7%: Germany 0.28%,[14] USA 0.42%,[14] UK 0.62%,[12] Australia 1.01%,[14] UK based on modelling 0.91%–1.62%,[15] Denmark 1.7%.[16]

Our estimates of AF-related costs in the large, prospective Swiss-AF cohort were highly consistent and robust. In particular, the regression-based estimates of AF costs using a matched control population were remarkably similar to the cost estimates based on direct adjudication to AF. The adjudication algorithm was derived using clinical and claims data for the Swiss AF sample only, without comparison to the population-based controls. So far, most literature has focused on estimating costs from clinical or claims data[3 4 6 9 10 30]; only very few comparisons with a control population are available.[8 16] While lending strong credibility to our results, the observed similarity also suggests that lacking controls, the AF-related portion of healthcare costs may still be estimated quite accurately with a well-defined algorithm supported by clinical data.

There are still several limitations of our work requiring discussion. Most importantly, the Swiss-AF study population is not truly representative of all AF patients in Switzerland, given enrolment in inpatient and outpatient clinical centres and an expected under-representation of patients younger than 65 years driven by eligibility criteria. It would in fact be extremely difficult, if not impossible, to recruit a truly representative sample of AF patients into any study. We still expect our cost estimates to provide a reasonable approximation of the typical AF-related costs of Swiss patients with clinically diagnosed AF. The decision to enrol patients independently of time since diagnosis supports this notion, all the more given the observed high degree of stability of our results over time. However, we cannot exclude that enrolment of the Swiss-AF patients in clinical centres may have led to a certain overestimation of inpatient cost in the first year of observation. Second, the selection algorithm used to define the control population is likely to have missed some patients with AF. However, the lack of exclusion of these patients should not have biased the results strongly, as they did not display indicators of AF-related hospitalisation or major procedures. If anything, a moderate underestimation of AF costs may have occurred. Third, cost calculations were based on claims data, and not all claims may have been handed in for reimbursement. However, in patients with a chronic disease and substantial healthcare costs, this is rather not expected. We could not acquire insurance characteristics from one insurer and have considered these in a sensitivity analysis without distortion of our results. Fourth, the controls were provided by one health insurance only. Major differences between insurers are not expected in the Swiss statutory health insurance, as the primary benefit package is uniform across the country and defined by law. A further limitation affects the estimation of the cost of illness at the national level. There were several assumptions made: (A) AF-related cost estimates were based on the results of the GLM-based two-part model, (B) the development of costs per patient over time was assumed to follow the development of healthcare expenditures in Switzerland and (C) AF patients under the age of 30 were not considered in the prevalence estimates. As a last limitation, this analysis focused on direct medical costs from the perspective of the Swiss statutory health insurance. Costs of lost productivity were not considered and the total impact of AF on the economy was thus not captured. Separate work will address the topic of impact of AF on productivity in younger Swiss-AF patients.

In conclusion, the results of this study indicate that AF patients incur 50% higher costs than comparable population-based controls. Costs were at a comparable level as reported by other cost-of-illness studies for AF. Different regression-based approaches to estimating AF-related costs led to similar results, confirming the robustness of our findings. A well-defined bottom-up approach using clinical and claims data but no control population also yielded similar results. This finding is valuable for the interpretation of the existing cost-of-illness literature and may inform decisions on investments in healthcare policies. To control the high costs of AF, future steps may include conducting real-world analyses to understand contributing factors and services, assessing the cost-effectiveness of AF-related treatments to guide resource allocation, and studying risk factors to develop targeted interventions aimed at reducing AF incidence and improving healthcare efficiency.

**Author affiliations**
[1]Epidemiology, Biostatistics and Prevention Institute, University of Zurich, Zurich, Switzerland
[2]Department of Health Sciences, Helsana Group, Zurich, Switzerland
[3]Department of Medicine, Cardiology Division, University Hospital Basel, Basel, Switzerland
[4]Cardiovascular Research Institute Basel, University Hospital Basel, Basel, Switzerland
[5]Department of Medicine, Baden Cantonal Hospital, Baden, Switzerland
[6]Center for Molecular Cardiology, University of Zurich, Schlieren, Switzerland
[7]Institute of Primary Care, University of Zurich, Zurich, Switzerland
[8]Institute of Primary Health Care (BIHAM), University of Bern, Bern, Switzerland
[9]Department of General Internal Medicine, Inselspital, University Hospital Bern, Bern, Switzerland
[10]Department of Neurology, University Hospital Basel, Basel, Switzerland
[11]Department of Research, Reha Rheinfelden, Rheinfelden, Switzerland
[12]Population Health Research Institute, McMaster University, Hamilton, Ontario, Canada
[13]Division of Cardiology, Cardiocentro Ticino (CCT), Lugano, Switzerland
[14]Faculty of Business and Economics, University of Basel, Basel, Switzerland
[15]Division of Cardiology, Ente Ospedaliero Cantonale (EOC), Instituto Cardiocentro Ticino, Ospedale Regionale di Lugano, Lugano, Switzerland
[16]Department of Cardiology, Triemli Hospital Zurich, Zurich, Switzerland
[17]Department of Cardiology, Inselspital, University Hospital Bern, Bern, Switzerland

[18]Institute of Pharmaceutical Medicine (ECPM), University of Basel, Basel, Switzerland

**Acknowledgements** We thank Lukas Kauer from the health insurance CSS, Pascal Godet from the health insurance Sanitas, Andri Signorell from the health insurance Helsana, and the health insurance KPT for providing the claims data for this study.

**Collaborators** Swiss-AF investigators University Hospital Basel and Basel University: Stefanie Aeschbacher, Katalin Bhend, Steffen Blum, Leo Bonati, David Conen, Ceylan Eken, Urs Fischer, Corinne Girroy, Elisa Hennings, Elena Herber, Vasco Iten, Philipp Krisai, Michael Kühne, Maurin Lampart, Mirko Lischer, Nina Mäder, Christine Meyer-Zürn, Pascal Meyre, Andreas U. Monsch, Luke Mosher, Christian Müller, Stefan Osswald, Rebecca E. Paladini, Anne Springer, Christian Sticherling, Thomas Szucs, Gian Völlmin. Principal Investigator: Stefan Osswald; Local Principal Investigator: Michael Kühne  University Hospital Bern: Faculty: Faculty: Drahomir Aujesky, Juerg Fuhrer, Laurent Roten, Simon Jung, Heinrich Mattle; Research fellows: Seraina Netzer, Luise Adam, Carole Elodie Aubert, Martin Feller, Axel Loewe, Elisavet Moutzouri, Claudio Schneider; Study nurses: Tanja Flückiger, Cindy Groen, Lukas Ehrsam, Sven Hellrigl, Alexandra Nuoffer, Damiana Rakovic, Nathalie Schwab, Rylana Wenger, Tu Hanh Zarrabi Saffari. Local Principal Investigator: Nicolas Rodondi, Tobias Reichlin  Stadtspital Triemli Zurich: Christopher Beynon, Roger Dillier, Michèle Deubelbeiss, Franz Eberli, Christine Franzini, Isabel Juchli, Claudia Liedtke, Samira Murugiah, Jacqueline Nadler, Thayze Obst, Jasmin Roth, Fiona Schlomowitsch, Xiaoye Schneider, Katrin Studerus, Noreen Tynan, Dominik Weishaupt. Local Principal Investigator: Andreas Müller  Kantonspital Baden: Simone Fontana, Corinne Friedli, Silke Kuest, Karin Scheuch, Denise Hischier, Nicole Bonetti, Alexandra Grau, Jonas Villinger, Eva Laube, Philipp Baumgartner, Mark Filipovic, Marcel Frick, Giulia Montrasio, Stefanie Leuenberger, Franziska Rutz. Local Principal Investigator: Jürg-Hans Beer  Cardiocentro Lugano: Angelo Auricchio, Adriana Anesini, Cristina Camporini, Maria Luce Caputo, Francois Regoli, Martina Ronchi. Local Principal Investigator: Giulio Conte  Kantonsspital St. Gallen: Roman Brenner, David Altmann, Michaela Gemperle. Local Principal Investigator: Peter Ammann  Hôpital Cantonal Fribourg: Mathieu Firmann, Sandrine Foucras, Martine Rime. Local Principal Investigator: Daniel Hayoz  Luzerner Kantonsspital: Benjamin Berte, Kathrin Bühler, Virgina Justi, Frauke Kellner-Weldon, Melanie Koch, Brigitta Mehmann, Sonja Meier, Myriam Roth, Andrea Ruckli-Kaeppeli, Ian Russi, Kai Schmidt, Mabelle Young, Melanie Zbinden. Local Principal Investigator: Richard Kobza  Ente Ospedaliero Cantonale Lugano: Elia Rigamonti, Carlo Cereda, Alessandro Cianfoni, Maria Luisa De Perna, Jane Frangi-Kultalahti, Patrizia Assunta Mayer Melchiorre, Anica Pin,Tatiana Terrot, Luisa Vicari. Local Principal Investigator: Giorgio Moschovitis.  University Hospital Geneva: Georg Ehret, Hervé Gallet, Elise Guillermet, Francois Lazeyras, Karl-Olof Lovblad, Patrick Perret, Philippe Tavel, Cheryl Teres. Local Principal Investigator: Dipen Shah  University Hospital Lausanne: Nathalie Lauriers, Marie Méan, Sandrine Salzmann, Jürg Schläpfer. Local Principal Investigator: Alessandra Pia Porretta  Bürgerspital Solothurn: Andrea Grêt, Jan Novak, Sandra Vitelli. Local Principal Investigator: Frank-Peter Stephan  Ente Ospedaliero Cantonale Bellinzona: Jane Frangi-Kultalahti, Augusto Gallino, Luisa Vicari. Local Principal Investigator: Marcello Di Valentino University of Zurich/University Hospital Zurich: Helena Aebersold, Fabienne Foster, Matthias Schwenkglenks.  Medical Image Analysis Center AG Basel: Jens Würfel (Head), Anna Altermatt, Michael Amann, Marco Düring, Petra Huber, Esther Ruberte, Tim Sinnecker, Vanessa Zuber.  Clinical Trial Unit Basel: Michael Coslovsky (Head), Pascal Benkert, Gilles Dutilh, Milica Markovic, Pia Neuschwander, Patrick Simon, Olivia Wunderlin Schiller AG Baar: Ramun Schmid

**Contributors** All authors certify that they have participated sufficiently in the work to take public responsibility for the content, including participation in the concept, design, analysis, writing, or revision of the manuscript. Concept: HA, FF-W, MS-B and MS. Design and methods: HA, FF-W, MS-B, MS, BB, SA, J-HB, EB, MB, LB, DC, GC, SF, CH, MK, GM, AM, REP, TR, NR, ASp, ASt, CS, TS and SO. Acquisition of data: HA, FF-W, MS-B, MS, BB, SA, ASp, MK, GM, REP and SO. Analysis: HA, FF-W, MS-B and MS. Interpretation of results: all authors. Writing: HA, FF-W, MS-B and MS. Reviewing it critically for important intellectual content: all authors. Final approval: all authors. Guarantor of this manuscript: HA and MS.

**Funding** This work is supported by grants of the Swiss National Science Foundation (grant numbers 105318_189195 / 1, 33CS30_148474, 33CS30_177520, 32473B_176178, and 32003B_197524), the Swiss Heart Foundation, the Foundation for Cardiovascular Research Basel (FCVR) and the University of Basel.

**Competing interests** Professor Beer reports grant support from the Swiss National Foundation of Science, The Swiss Heart Foundation and the Stiftung Kardio; grant support, speakers- and consultation fees to the institution from Bayer, Sanofi and Daichii Sankyo. Dr Blozik reports grants from Swiss Cancer Research Foundation; institutional grants from Amgen, MSD, Novartis, Pfizer, all outside the submitted work. Dr Bonati reports personal fees and nonfinancial support from Amgen, grants from AstraZeneca, personal fees and nonfinancial support from Bayer, personal fees from Bristol-Myers Squibb, personal fees from Claret Medical, grants from Swiss National Science Foundation, grants from University of Basel, grants from Swiss Heart Foundation, outside the submitted work. Dr Conen received consulting fees from Roche Diagnostics, and speaker fees from Servier and BMS/Pfizer, all outside of the current work. Dr Kühne reports personal fees from Bayer, personal fees from Böhringer Ingelheim, personal fees from Pfizer BMS, personal fees from Daiichi Sankyo, personal fees from Medtronic, personal fees from Biotronik, personal fees from Boston Scientific, personal fees from Johnson&Johnson, personal fees from Roche, grants from Bayer, grants from Pfizer, grants from Boston Scientific, grants from BMS, grants from Biotronik, grants from Daiichi Sankyo. Dr Moschovitis has received consultant fees for taking part to advisory boards from Novartis, Boehringer Ingelheim, Bayer, Astra Zeneca and Daiichi Sankyo, all outside of the presented work. Dr Müller reports fellowship and training support from Biotronik, Boston Scientific, Medtronic, Abbott/St. Jude Medical, and Biosense Webster; speaker honoraria from Biosense Webster, Medtronic, Abbott/ St. Jude Medical, AstraZeneca, Daiichi Sankyo, Biotronik, MicroPort, Novartis, and consultant honoraria for Biosense Webster, Medtronic, Abbott/St. Jude Medcal, and Biotronik. Dr Osswald received research grants from the Swiss National Science Foundation and from the Swiss Heart Foundation, research grants from Foundation for CardioVascular Research Basel, research grants from Roche, educational and speaker office grants from Roche, Bayer, Novartis, Sanofi AstraZeneca, Daiichi-Sankyo and Pfizer. Dr Reichlin has received research grants from the Swiss National Science Foundation, the Swiss Heart Foundation, and the sitem insel support fund, all for work outside the submitted study. Speaker/consulting honoraria or travel support from Abbott/SJM, Astra Zeneca, Brahms, Bayer, Biosense-Webster, Biotronik, Boston-Scientific, Daiichi Sankyo, Medtronic, Pfizer-BMS and Roche, all for work outside the submitted study. Support for his institution's fellowship program from Abbott/SJM, Biosense-Webster, Biotronik, Boston-Scientific and Medtronic for work outside the submitted study. Dr Schwenkglenks reports grants from Swiss National Science Foundation, for the conduct of the study; grants and personal fees from Amgen, grants from MSD, grants from Novartis, grants from Pfizer, grants from The Medicines Company, all outside the submitted work. Dr Serra-Burriel reports grants from the European Commission outside of the present work. Dr Sticherling has received speaker honoraria from Biosense Webster and Medtronic and research grants from Biosense Webster, Daiichi-Sankyo, and Medtronic. The remaining authors have nothing to disclose.

**Patient and public involvement** Patients and/or the public were not involved in the design, or conduct, or reporting, or dissemination plans of this research.

**Patient consent for publication** Not applicable.

**Ethics approval** The Swiss-AF study protocol was approved by the local ethics committee (Ethikkommission Nordwest- und Zentralschweiz, 2014-067, PB_2016-00793), and written informed consent was obtained from each participant. The population-based reference data set from Helsana was provided anonymously based on a waiver provided by the competent ethics committee (Kantonale Ethikkommission Zürich, 2020-01346).

**Provenance and peer review** Not commissioned; externally peer reviewed.

**Data availability statement** Data are available on reasonable request. Data may be obtained from a third party and are not publicly available. The Swiss-AF patient informed consent forms state that the data, containing personal and medical information, are exclusively available for research institutions in an anonymised form and are not allowed to be made publicly available. Researchers interested in obtaining the Swiss-AF data for research purposes can contact the Swiss-AF scientific lead. Contact information is provided on the Swiss-AF website (http://www.swissaf.ch/contact.htm). Authorisation of the responsible ethics committee is mandatory before the requested data can be transferred to external research institutions.The population-based reference data set from Helsana was provided anonymously. These claims data cannot be shared publicly because they are the property of Helsana. Considering SNSF policies encouraging data sharing, the data may be shared via Helsana with scientific institutions under specific conditions and considering all data protection rules; final decisions are taken by Helsana. Any such sharing would also require ethical clarification of responsibility and/or clearance, as applicable.

**ORCID iDs**
Helena Aebersold http://orcid.org/0000-0002-4418-9904
Beat Brüngger http://orcid.org/0000-0001-6173-5375
Carola Huber http://orcid.org/0000-0002-2469-0435
Giorgio Moschovitis http://orcid.org/0000-0002-4043-8061

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
