## [Reviewer comments · BMJ Open]

ARTICLE DETAILS

TITLE (PROVISIONAL)	Estimating the cost impact of atrial fibrillation using a prospective cohort study and population-based controls
AUTHORS	Aebersold, Helena; Foster-Witassek, Fabienne; Serra-Burriel, Miquel; Brüngger, Beat; Aeschbacher, Stefanie; Beer, Jürg-Hans; Blozik, Eva; blum, manuel; Bonati, Leo; Conen, David; Conte, Giulio; Felder, Stefan; Huber, Carola; Kuehne, Michael; Moschovitis, Giorgio; Mueller, Andreas; Paladini, Rebecca E; Reichlin, Tobias; Rodondi, Nicolas; Springer, Anne; Stauber, Annina; Sticherling, Christian; Szucs, Thomas; Osswald, Stefan; Schwenkglenks, M

VERSION 1 – REVIEW

REVIEWER	Faizel Osman University Hospitals Coventry and Warwickshire NHS Trust, Cardiology
REVIEW RETURNED	27-Mar-2023

GENERAL COMMENTS	This is an interesting and helpful paper evaluating a large cohort of Swiss patients looking at the cost implications of atrial fibrillation in a large population-based cohort. The paper is well written and presented. The detailed statistical methods used requires further expert statistical input which I have not been able to provide in this review. I have the following comments/minor points: 1. Why was the time frame 2014-2017 chosen? Clearly the impact of the COVID-19 pandemic will have had an effect on the findings. A more up-to-date analysis may have provided a more contemporaneous data-set. However, the data presented are still very interesting and consistent with other data that are available.2. The AF cohort was captured from insurance company covering 42% of all AF patients. This is clearly a limitation and has been appropriately acknowledged by the authors. This clearly may have impact on interpretation of the data set.3. Page 6, line 52: The data is presented as 'AF likely', 'AF possible' and, 'AF not obvious'. The authors use the term severe AF. This is a term that is not accurate as severity is not being assessed. I would suggest the authors change this to something more appropriate, such as probable/definite AF rather than severe AF.4. Page 6, line 47: they mention the term Swiss DRG. Can the authors clarify what DRG stands for.5. Page 7, line 50; the authors use the word 'acid related disorders.' This should be clarified as gastro-oesophageal reflux disease. This should also be clarified in Table 1 on page 10.6. The authors use SMD in Table 1 (page 10) rather than p-values or
---

	confidence intervals for comparing AF versus control patients. It is unclear whether this metric is an assessment of significance or not and I would suggest specialised statistical input for this and other statistics used. 7. Table 1, page 10: The authors describe 'mother tongue' and put it under the heading 'socio-economic'. However primary language does not indicate socioeconomic status and the authors should clarify this in the table and text.
--	--

REVIEWER	Nipun Atreja Bristol-Myers Squibb Company
REVIEW RETURNED	05-Jun-2023

GENERAL COMMENTS	Dear Author, Thanks for the opportunity to review this manuscript. The study aimed to estimate the impact of Atrial Fibrillation on direct healthcare costs in a Swiss setting. This is a well conducted and nicely written manuscript. Please find my comments and suggestions below Introduction: There is plenty of cost data for AF including for AF subgroups in the Europe as well as in US. Can the authors strengthen the rationale as to why do we need to compare AF cost to a control cohort of non-AF patients as the two populations would be different. Additionally, AF patients suffer from significant comorbidities which cause significant cost burden, and these comorbidities are similar across the control group. Methods: Random selection of non-AF can introduce bias. A matched approach can provide better estimate than random assignments for start and end date. Also, it does not clear how the "Individuals within the reference sample could have AF" were treated as this can bias the findings for the reference cohort. Covariates: Do the listed covariates cover majority of comorbidities for AF and reference cohort? Comorbidities should be different for the two cohorts. Discussion: Suggest adding implications of high direct AF cost and any future steps to control the cost associated with AF.
---

VERSION 1 – AUTHOR RESPONSE

Reviewer 1 (7 issues):

Comment 1: Why was the time frame 2014-2017 chosen? Clearly the impact of the COVID-19 pandemic will have had an effect on the findings. A more up-to-date analysis may have provided a more contemporaneous data-set. However, the data presented are still very interesting and consistent with other data that are available.

Response: Enrolment of Swiss-AF patients was 2014-2017, while the analysis used a data cut from 2014-2020. Individual patient follow-up was censored at five years after enrolment, given sparse data thereafter.

Changes to the manuscript: We have added this information to the methods section in the abstract to improve clarity, p. 3.

Comment 2: The AF cohort was captured from insurance company covering 42% of all AF patients. This is clearly a limitation and has been appropriately acknowledged by the authors. This clearly may have impact on interpretation of the data set.

Response: The availability of health insurance claims relied on the cooperation of health insurers. Specifically, four health insurers participated, covering 42% of the study sample. In Switzerland, health insurance is mandatory and accessible to everyone, providing coverage for both inpatient and outpatient services. The benefit package offered by health insurers is uniform nationwide and defined by law. Significant variations between insurers are therefore not expected. Patients with and without available claims data in Swiss-AF had comparable characteristics (Table S1 in Aebersold et al., 2022)¹.

Changes to the manuscript: None.

Comment 3: Page 6, line 52: The data is presented as 'AF likely', 'AF possible' and, 'AF not obvious'. The authors use the term severe AF. This is a term that is not accurate as severity is not being assessed. I would suggest the authors change this to something more appropriate, such as probable/definite AF rather than severe AF.

Response: Thank you, we have changed the wording in the manuscript.

Changes to the manuscript: p. 7

Comment 4: Page 6, line 47: they mention the term Swiss DRG. Can the authors clarify what DRG stands for.

Response: DRG stands for diagnosis related group. We have moved the abbreviation to the front of the sentence, to make clearer where it belongs to.

Changes to the manuscript: p. 7

Comment 5: Page 7, line 50; the authors use the word 'acid related disorders.' This should be clarified as gastro-oesophageal reflux disease. This should also be clarified in Table 1 on page 10.

Response: We have followed the terminology of the pharmaceutical cost group algorithm used by Huber et al., 2013.² However, to improve clarity, we have added "gastro-oesophageal reflux dis-ease" to the manuscript text, in brackets.

Changes to the manuscript: p. 8

Comment 6: The authors use SMD in Table 1 (page 10) rather than p-values or confidence intervals for comparing AF versus control patients. It is unclear whether this metric is an assessment of significance or not and I would suggest specialised statistical input for this and other statistics used.

Response: Usually, the p-value is calculated to decide whether two conditions, e.g. control and treatment, are statistically different. The occurrence of significant p-values is partially a function of sample size. Also, p-values cannot quantify the magnitude of differences, failing to answer a very relevant question: "How large is the difference between the conditions?"^{3,4}

The standardized (mean) difference quantifies the distance between two group means, based on one or more variables. In practice it is commonly used as an indicator of balance for individual co-variables before and after propensity score matching.⁵

Changes to the manuscript: Currently, none. We had originally included p-values but decided to remove them for the above-stated reasons. However, if the Editorial office prefers this, we are happy to re-include them.

Comment 7: Table 1, page 10: The authors describe 'mother tongue' and put it under the heading 'socio-economic'. However primary language does not indicate socioeconomic status and the authors should clarify this in the table and text.

Response: Thank you, we have changed the heading accordingly.

Changes to the manuscript: p. 11

Reviewer 2 (6 issues):

Comment 1: Introduction: There is plenty of cost data for AF including for AF subgroups in the Europe as well as in US. Can the authors strengthen the rationale as to why do we need to compare AF cost to a control cohort of non-AF patients as the two populations would be different.

Response: To isolate the effects of a disease, the comparison against a disease-free control group is a well-established method for conducting cost-of-illness analyses, especially when dealing with large sample sizes.^{6,7} This approach has been widely used in research, and it helps mitigate the uncertainties associated with adjudicating medical resource usage to a specific disease of interest. However, as we could show in the present article, such adjudication can work well in the case of AF. Changes to the manuscript: None.

Comment 2: Additionally, AF patients suffer from significant comorbidities which cause significant cost burden, and these comorbidities are similar across the control group.

Response: The control group was matched to the AF group for age, gender and geographic region of living. The similarity observed for major non-cardiovascular disease categories occurred 'natural-ly'. In our interpretation, this supports the suitability of our approach to isolate the costs of AF and its complications, despite limitations, which we discuss.

Changes to the manuscript: None.

Comment 3: Methods: Random selection of non-AF can introduce bias. A matched approach can provide better estimate than random assignments for start and end date.

Response: Random selection was for the control population from the general population. The aim was to provide a representative sample but matched to the AF group (see response to Reviewer 2, Comment 2). The start dates of the observation periods for the controls were assigned using the distribution of Swiss-AF enrolment to ensure similar calendar time periods being considered. However, the end dates were not randomly assigned, since events were considered as they occurred, with a censoring after 5 years of follow-up, to ensure a similar observation period as in the Swiss-AF population. To confirm this did not lead to distortions, sensitivity analyses with different starting as well as different ending dates were conducted, as stated in the manuscript. No significant changes to the results were seen.

Changes to the manuscript: None.

Comment 4: Also, it does not clear how the "Individuals within the reference sample could have AF" were treated as this can bias the findings for the reference cohort.

Response: In the last paragraph on p. 7 of the manuscript, we explain how potential controls were categorized as "AF likely", "AF possible", and "AF not obvious", based on a categorization algorithm developed together with clinicians. Only persons with "AF not obvious" were finally considered as controls (see end of page 7). In the discussion (last paragraph on p. 16), we address and judge the risk of distortions potentially emerging from the fact that we could not safely exclude all patients with AF.

Changes to the manuscript: To add clarity, we have amended the sentence in the discussion, p. 16.

Comment 5: Covariates: Do the listed covariates cover majority of comorbidities for AF and reference cohort? Comorbidities should be different for the two cohorts.

Response: Adding to our response to Reviewer 2, Comment 2, it may be relevant to reiterate that the available covariates representing comorbidities were based on pharmaceutical cost groups, as they can be derived from drug claims in health insurance claims data. We cannot compare specifically AF-related conditions/complications, because we only have clinical data for the AF patients. Here, we would of course expect differences.

Changes to the manuscript: None.

Comment 6: Discussion: Suggest adding implications of high direct AF cost and any future steps to control the cost associated with AF.

Response: Thank you for this valuable input, we have added some implications to the discussion.

Changes to the manuscript: p. 17